# Navigating Perceived Stress: Experiences of Nursing Students Completing Internships during the COVID-19 Pandemic in Spain

**DOI:** 10.3390/jcm13164943

**Published:** 2024-08-22

**Authors:** María-Ángeles Merino-Godoy, Zaida Montero Aceijas, María Cano Martín, Francisco-Javier Gago-Valiente, Alberto Vega Abengozar, Juan María Pérez Padilla, Emilia Teixeira da Costa

**Affiliations:** 1Nursing Department, Faculty of Nursing, University of Huelva, 21007 Huelva, Spain; angeles.merino@denf.uhu.es; 2Institute Pere Mata, University Psychiatric Hospital, 43206 Tarragona, Spain; monteroz@peremata.com; 3Osuna Hospital, 41640 Seville, Spain; maria.cano.martin.sspa@juntadeandalucia.es; 4Center for Research in Contemporary Thought and Innovation for Social Development (COIDESO), University of Huelva, 21007 Huelva, Spain; 5Los Ángeles De La Salud, 21600 Huelva, Spain; info@losangelesdelasalud.es; 6Erican Rescue ONG, 41008 Seville, Spain; ericanrescate@ericanrescate.es; 7Nursing Department, Health School, University of Algarve, 8000 Faro, Portugal; eicosta@ualg.pt; 8Health Sciences Research Unit: Nursing, Nursing School of Coimbra, 3000 Coimbra, Portugal

**Keywords:** nursing students, stress, mental health, coronavirus, public health crises

## Abstract

**Background:** University students often experience psychological strains such as academic stress, particularly as they approach the transition into the workforce. This stress may have been heightened for nursing students who completed internships during the COVID-19 pandemic. This study aims to analyze the impact of the COVID-19 pandemic on the perceived stress levels of undergraduate nursing students. **Methodology:** A cross-sectional descriptive observational study was conducted using the Spanish version of the PSS-10 scale, a validated reduction of the English version PSS-14, to evaluate perceived stress. The responses are Likert-type with a total score range of 0 to 40. Questionnaires were distributed electronically to nursing students across all academic years who were engaged in clinical practice. Participation was voluntary. **Results:** The study included 487 students, the majority of whom were women (78.4%) with an average age of 23.51 years. Most participants were in their third and fourth years (67%). The mean perceived stress score was 20.65 (SD = 5.62) out of a possible 40, indicating moderate stress levels. Statistically significant differences in perceived stress were found between genders, with women reporting higher stress levels than men (Mann–Whitney U = 15,380.000; *p* < 0.001). Additionally, a significant correlation was observed between the overall perceived stress score and gender, as well as between specific items on the PSS-10 scale and gender, highlighting the importance of gender-specific stress management interventions. **Conclusions:** Nursing students reported moderate levels of perceived stress, with women experiencing higher stress levels than men. These findings highlight the need for targeted stress management interventions for nursing students, especially during health crises. Addressing gender-specific stressors and fostering a supportive educational environment will enhance students’ well-being, academic success, and professional preparedness.

## 1. Introduction

The emergence of COVID-19 in Wuhan, China, in December 2019 marked the beginning of a global crisis that swiftly transformed daily life across the world [1]. This pandemic profoundly impacted societies [2]. As the SARS-CoV-2 virus led to escalating numbers of infections and fatalities, the World Health Organization (WHO) officially designated it a pandemic on 20 January 2020 [3]. The ensuing global lockdowns drastically altered people’s everyday activities [4,5].

On 14 March 2020, the Spanish government implemented a national lockdown as a preventive measure against COVID-19. This strategy encompassed various actions, including restrictions on mobility, emergency investments in health centers, and the closure of educational institutions. These measures significantly impacted the mental health of the population, leading to a rise in psychological issues such as anxiety, depression, and stress [6].

University education was significantly affected by the pandemic measures. Institutions shifted to e-learning, utilizing digital resources and platforms to maintain educational continuity [7,8]. However, the limited experience and capacity of both teachers and students to use these new platforms presented several challenges [9,10]. Additionally, the lack of consideration for students’ access to the internet led to heightened inequality, further exacerbating the crisis [7]. Over 1.5 billion students were impacted by the closure of schools and universities worldwide [11]. This disruption also negatively affected university students’ mental health, resulting in high levels of stress, increased blood pressure, insomnia, and weakened immune systems [3].

The shift to online learning was a novel experience for students, leading to confusion about developing their personal competences and skills. Social interactions were significantly reduced to a virtual environment, which negatively impacted their emotional well-being [3,12]. 

Moreover, online classes not only led to a loss of interaction with classmates and teachers but also disrupted clinical practice internships. This disruption was particularly detrimental for nursing students, whose training heavily relies on hands-on clinical experience. Consequently, their anxiety and stress levels increased significantly [13]. 

Spain experienced very high infection rates and a significant proportion of deaths due to COVID-19, with health professionals being among the most affected groups. These professionals worked under harsh conditions for extended periods, with limited personal protective equipment, and cared for highly complex patients [14,15]. This stressful situation increased their risk of anxiety, depression, and post-traumatic stress disorder [14]. Consequently, many developed compassion’s fatigue [16]. This syndrome likely indirectly affected trainees, who had to assume more responsibilities to support the nursing staff during the pandemic [15].

University students, particularly those in rigorous programs like nursing, typically experience significant psychological stressors related to academic demands, clinical training, and the transition from theoretical learning to hands-on practice. Before the COVID-19 pandemic, these stressors for nursing students primarily involved managing time, preparing for exams, and coping with the pressure of providing competent care during clinical placements. The challenges of integrating into healthcare settings as novices, coupled with the high expectations placed upon them, were central to their stress experiences [17].

However, the onset of the COVID-19 pandemic drastically altered the nature and intensity of these stressors. Nursing students suddenly found themselves at the frontline of a global health crisis, facing new and more complex stressors that compounded their existing challenges. The pandemic introduced significant health risks, with students fearing for their own safety and the possibility of transmitting the virus to their families. Clinical training was disrupted, leading to concerns about the adequacy of their practical skills. Additionally, many students were required to take on increased responsibilities during their placements due to staff shortages, all while dealing with the emotional toll of the pandemic, including isolation and uncertainty about their futures [13].

Research during the COVID-19 pandemic has shown significant stress variations by age and gender. Younger students often reported higher stress due to less developed coping mechanisms and the sudden shift to remote learning, while older students, with more life experience, reported lower stress levels. Additionally, female students generally experienced higher stress than males, likely due to gender-specific factors. Understanding these variations is crucial for analyzing how different demographic groups, particularly in nursing, were affected during the pandemic [13].

This situation was exceptional for all university programs in general and particularly challenging for nursing degrees. The rapid shift to virtual classes, the suspension of some clinical practice periods, and the increased responsibilities in subsequent clinical placements created significant challenges. These factors may have led students to question whether they had acquired the necessary competencies and practical skills to complete their degree [18]. Therefore, this research aims to analyze the perceived stress experienced by nursing students during their clinical placements amidst the COVID-19 pandemic. 

## 2. Materials and Methods

### 2.1. Study Design and Participants

A cross-sectional, descriptive, observational study was conducted with 487 undergraduate nursing students from various regions of Spain. 

The fieldwork for this study was conducted in March and April 2022, a period during which the immediate effects of the COVID-19 pandemic had been somewhat mitigated due to widespread vaccination campaigns. However, the mandatory use of face masks in health centers, hospitals, pharmacies, and socio-health centers across Spain was still in effect, as mandated by the Spanish government. The official end of this mandate, which marked the conclusion of the health crisis, did not occur until 5 July 2023, when Order SND/726/2023 was published in the Official State Gazette (BOE). This context is crucial as it highlights that nursing students participating in clinical practice during the study period were still operating under significant precautionary measures and were exposed to ongoing risks associated with COVID-19. These conditions likely contributed to the elevated levels of perceived stress observed among the participants, making the reference to health crises in our study’s conclusions both relevant and necessary.

A comprehensive work schedule was created, detailing specific timelines for each phase of the research. To participate in this study, students needed to meet specific inclusion criteria: they had to be currently enrolled in a nursing degree program at a university, be actively involved in or have recently completed a clinical placement during the COVID-19 pandemic, and voluntarily agree to participate by providing informed consent. Students were excluded from the study if they were not enrolled in a nursing degree program, had not participated in a clinical placement during the COVID-19 pandemic, did not provide informed consent, or had already completed their nursing degree before the pandemic began.

To recruit participants, a multi-faceted approach was employed. Students from various universities across Spain were contacted through multiple channels, including social networks (Instagram, Twitter, WhatsApp), email, and telephone. Additionally, student representatives from different universities were enlisted to aid in the dissemination of the questionnaires. These representatives were responsible for spreading the word to students from all years of the nursing program, encouraging voluntary participation.

The questionnaires were developed using Google Forms, a user-friendly online survey tool, ensuring ease of access and completion. Data collection was facilitated directly on this platform, streamlining the process and ensuring accurate data capture.

Before starting the questionnaire, participants were required to provide informed consent, which was prominently featured at the beginning of the form. This step was crucial to guarantee participants’ anonymity and ensure ethical compliance. The study received approval from the Huelva Research Ethics Committee, with the general code AMG-COV-2021-02B and the internal code 0797-N-22, ensuring that all ethical standards were upheld throughout the research process.

### 2.2. Instruments

Participants were asked to complete a brief socio-demographic questionnaire designed to gather essential background information. This questionnaire collected data on various factors, including their age, gender, marital status, living arrangements (such as living alone, with family, with a partner, or with flatmates), and current academic year. Moreover, to gain insight into the impact of the pandemic on their educational choices, an item at the end of the questionnaire specifically inquired whether participants had considered dropping out of their nursing studies as a result of their experiences during the pandemic. This additional question aimed to understand the extent to which the pandemic situation influenced their commitment to continuing their nursing education. Additionally, participants were asked whether they had ever contracted COVID-19, whether they had received specific training on protection against this issue, and how they experienced close contact with the disease.

Perceived stress was measured using the perceived stress scale (PSS-10). This scale comprises 10 questions designed to identify the perceived stress levels of respondents by exploring various stressors. The Spanish version, as utilized by Remor [19], was employed for this study. The PSS-10 measures the extent to which individuals perceive everyday life situations as stressful. It consists of direct questions that assess stress experienced over the past month, with responses provided on a Likert scale. While the PSS-10 is not specific to COVID-19, it captures general stress levels, which are relevant to understanding how nursing students perceived stress in a high-pressure environment exacerbated by the pandemic. The decision to use the PSS-10 was based on its strong psychometric properties, including reliability and validity in diverse populations, including university students.

The scale includes 10 items with response options ranging from “never” to “always”, scored from 0 to 4. Items 1, 2, 3, 6, 9, and 10 are scored directly, while items 4, 5, 7, and 8 are reverse-scored [20]. Thus, higher overall scores indicate higher perceived stress levels. The possible scores range from 0 to 40, categorized into three levels: low (0–10), moderate (11–25), and high (26–40) [21].

The PSS-10 has been validated with university student populations and demonstrates good internal consistency. Its validity was confirmed through exploratory and confirmatory factor analyses. Internal consistency was measured with Cronbach’s alpha (0.859) and McDonald’s omega (0.887) coefficients, indicating robust reliability [22].

### 2.3. Data Analysis

The analysis of the variables was conducted using the Statistical Package for the Social Sciences (SPSS) version 27.0. The reliability of the Spanish-translated PSS-10 scale was assessed using Cronbach’s alpha, which yielded a value above 0.70, indicating good reliability. A univariate analysis was performed to calculate the mean, standard deviation, minimum, and maximum values for the quantitative variables. Frequencies and percentages were also computed for categorical variables such as gender, marital status, academic year, PSS-10 results, place of accommodation during the school year, and the intention to drop out of the nursing bachelor’s degree.

Normality tests were applied to the quantitative variables to determine the appropriate hypothesis tests, whether parametric or non-parametric. Given that the sample size exceeded 50, the Kolmogorov–Smirnov test was selected for assessing normality.

Subsequently, statistical tests were conducted according to the objectives of the study. For instance, a cross-tabulation was performed for item 7 of the PSS-10 scale and gender, and Chi-Square tests were conducted to examine the relationship between these variables. Since the normality tests for the overall PSS-10 score indicated a non-normal distribution, the Mann–Whitney U test was utilized for independent two-category categorical variables. Additionally, Spearman’s Rho was applied to analyze correlations between various study variables, including the overall PSS-10 score in relation to gender, with the Mann–Whitney U test applied for hypothesis testing between these variables. The Spearman’s Rho test was also used to study the correlation between the overall perceived stress scale score and age.

This comprehensive analytical approach ensured the robustness and reliability of the study’s findings, providing valuable insights into the perceived stress among nursing students during their internships amid the COVID-19 pandemic in Spain.

## 3. Results

### 3.1. Socio-Demographic Characterization

A study was conducted with 487 undergraduate nursing students from various regions of Spain. Among the participants, 78.4% (*n* = 382) were women and 21.6% (*n* = 105) were men. The average age was 23.29 years (SD = 5.861). Marital status distribution was as follows: 4.7% married, 0.4% divorced, 85.8% single, and 9% reported other statuses. In terms of living arrangements during the school year, 6.6% lived alone, 49.1% with their family, 5.3% with their partner, and 39% with roommates. Of the participants, 255 were from the province of Andalusia and the remaining 232 were from other regions of Spain.

Most of the participating students were in the third year of the University Degree in Nursing (33.7%), and the lowest percentage of students were in the first year (7%). Only 9.4% of the people who collaborated in the research reported having the intention of abandoning the bachelor’s degree in nursing (Table 1).

Of the students surveyed, a significant majority, 68.8%, did not contract COVID-19. Conversely, a substantial portion, 80.3%, received training related to the use of protective equipment against COVID-19. The data further reveal that 85.1% of respondents feel uncomfortable when thinking about the disease, and 85.4% are concerned about contracting it. Additionally, 52.9% experience sweaty hands when considering the illness, and 79.8% fear dying from it. A high percentage, 81.8%, worry about potentially infecting family or friends. Furthermore, 51% report that COVID-19 causes them anxiety when leaving home to fulfill their obligations, while 15.7% feel nervous and 14.7% feel anxious about COVID-19.

### 3.2. Perceived Stress in the Participant Population

The mean perceived stress score among the participant population was 20.65 points. This indicates that, on average, the participants experienced a moderate level of stress. Furthermore, the median score, which was 21, confirms that half of the sample reported moderate stress levels, as reflected in their responses to the PSS-10 scale (Table 2). These findings highlight the prevalent moderate stress experienced by nursing students during the study period.

Table 3 presents the descriptive statistics, including frequency (n) and percentage (%), for each of the items on the PSS-10 scale. Notably, item 7, which inquiries about the frequency with which participants felt they could control difficulties in life, had the highest percentage of responses in the “almost never” category, at 42.7%.

### 3.3. Correlation of Item 7 of the PSS-10 Scale with Gender

In the analysis of the relationship between gender and the response option for item 7, significant differences were found (x^2^ = 11.640; *p* = 0.020 at a confidence level of 95%). This aspect is particularly noteworthy considering that the nursing profession continues to be heavily represented by women. Given the importance of this aspect in relation to nursing care practices, a deeper analysis was conducted. Specifically, a comparison of proportions between gender and responses to this item was performed, as detailed in Table 4. This analysis aims to shed light on potential gender differences in perceived control over life difficulties, which is crucial for understanding stress management among nursing students. The response option with the highest percentage for both men and women is “Rarely”. However, females had higher scores for this option.

### 3.4. Correlation of the Overall PSS-10 Score with Gender

When analyzing whether there were significant differences in the overall score of the PSS-10 scale according to gender, there was dependence between both variables (Mann–Whitney U = 15,380.000; *p* ≤ 0.001 at 95% confidence level). Women showed a higher mean range of perceived stress than men (Table 5).

### 3.5. Correlation of the Overall Score of the PSS-10 Scale with Age

When analyzing whether there was a dependence between the overall PSS-10 score and age, it was found that there were no statistically significant differences, that is, the age of the participants had no influence on the overall scale score (Table 6). 

## 4. Discussion

This study aimed to assess the stress levels experienced by undergraduate nursing students engaged in clinical practice during the COVID-19 pandemic. It is important to recognize that a pandemic can introduce numerous stressors in the workplace, potentially escalating factors that adversely affect the mental health of both workers and university students on clinical placements. In such circumstances, students may encounter negative emotions, heightened stress, and a reduction in their academic self-efficacy [23]. 

Several studies have investigated the perceived stress among students during the COVID-19 pandemic, providing a broader context for our findings. For example, some researchers have reported heightened stress levels among nursing students during the pandemic, highlighting the necessity for targeted interventions [13]. Additionally, other studies have found that stronger resilience and the use of humor were associated with significantly lower anxiety levels, while mental disengagement was linked to higher anxiety levels [24]. These conclusions are consistent with our study and emphasize the critical need for diverse and effective stress management strategies in educational environments. The results of this research corroborate these observations, revealing a moderate level of stress among the student population. This moderate stress and anxiety likely stemmed from various challenges related to the pandemic, including socio-economic, educational, and health issues. Furthermore, it is important to note that moderate stress often serves as a primary catalyst for emotional exhaustion [24]. 

Furthermore, the scientific literature reveals similar findings in other studies conducted on university health students. These studies indicate a significant impact on items evaluating personal capacities to manage stressful situations [25]. This trend is also evident in our study, particularly in the PSS-10 scale items related to personal capabilities. Some authors provide a possible explanation for this phenomenon [26]. They suggest that, when individuals perceive that external demands exceed their personal coping resources, these demands pose a threat and compromise their well-being. This understanding highlights the importance of assessing and enhancing personal coping mechanisms among nursing students to mitigate stress and improve their overall well-being.

In terms of personal coping with stress as a function of age, it is commonly assumed that older individuals may possess certain advantages that contribute to better stress management. These advantages may include greater emotional maturity, more developed anxiety management skills, and a broader range of life experiences that can provide valuable resources when facing stressful situations. Older students might have had more opportunities to develop resilience and adaptive coping mechanisms through previous life challenges, which theoretically would allow them to navigate stressful environments, such as clinical placements during a pandemic, more effectively than their younger counterparts [27].

However, in our study, no significant relationship was found between age and stress levels among the participants. This finding suggests that the expected benefits of age-related coping mechanisms did not manifest significantly within our sample. One possible explanation for this could be the relatively narrow age range of the participants, as the majority of nursing students are typically young adults with limited variance in age. The homogeneity of the age distribution in our sample may have minimized the observable differences in stress levels based on age. Furthermore, the unique and unprecedented nature of the COVID-19 pandemic could have introduced stressors that equally challenged all age groups, thereby diminishing the protective effects typically associated with older age.

It is also worth considering that the pandemic may have presented stressors that were novel and overwhelming, even for those who might otherwise cope well under normal circumstances. The rapid shift to virtual learning, the disruption of clinical training, and the ongoing fear of infection created a universally stressful environment, potentially leveling the playing field across different age groups. As a result, the expected differences in stress levels by age may not have emerged as strongly as they might have in a more stable and predictable context.

Given these findings, it is crucial for future research to explore the complex interplay between age, coping mechanisms, and stress in more varied and broader age groups. Additionally, longitudinal studies could provide deeper insights into how stress and coping evolve over time, particularly in response to prolonged crises like the COVID-19 pandemic. The scientific literature presents contradictory findings regarding the relationship between age and stress levels. Some studies have found a correlation between age and stress levels [28,29], while others have reported no statistically significant differences [30]. These inconsistencies highlight the need for further research to clarify the role of age in stress management, particularly within the context of nurses or nursing students.

Regarding the perception of stress levels by gender, this study found a significant association between the two variables. Women reported higher levels of perceived stress compared to their male counterparts. This finding is consistent with previous research, which also observed that female participants tend to experience higher levels of stress [31]. Several factors could contribute to this gender disparity in stress perception. Women may face unique stressors, including societal expectations, balancing multiple roles, and higher emotional labor, which can intensify their stress levels. Additionally, women might be more likely to report their stress or perceive situations as more stressful due to differences in coping mechanisms and socialization patterns [32].

These findings underscore the importance of addressing gender-specific stressors and developing targeted interventions to support female nursing students. By recognizing and mitigating these stress factors, educational institutions and healthcare settings can improve the overall well-being and academic success of their students. Further research is necessary to explore the underlying causes of this gender disparity and to develop effective strategies for stress management tailored to the needs of female students.

Nursing students frequently encounter various stressful situations during their training, such as feeling powerless to solve problems, experiencing work overload, or lacking expertise in care techniques. These stressors can lead to physical reactions like chronic fatigue or persistent tiredness [33]. Additionally, the pandemic has introduced further variables, including depression, anxiety, and sleep disorders [34]. These combined factors likely contributed to the moderate stress levels perceived by the participants in this study.

In addition to the challenges posed by the pandemic, the shift in teaching methodology and home confinement decreed by authorities significantly heightened anxiety and academic stress among students. Nursing students, who typically face high levels of stress due to factors such as time constraints, exam pressures, and heavy workloads, experienced an exacerbation of these issues during the pandemic [35,36]. The additional stress induced by COVID-19 likely had both internal and external repercussions, affecting their mental health, academic performance, and overall well-being [24]. These findings underscore the urgent need for targeted interventions and support systems to help nursing students manage stress, especially in the context of ongoing or future public health crises.

### 4.1. Limitations

Several limitations should be considered when interpreting the results of this study. Firstly, although the sample size was substantial and included students from various universities across Spain, variations in how these universities and associated health centers managed the pandemic could have influenced perceived stress levels. Some institutions may have implemented more effective coping strategies and support systems, resulting in lower stress levels among their students, while others may have been less effective, leading to higher stress levels.

Additionally, the study’s focus on nursing students in Spain during a specific time period may limit the generalizability of the findings to other populations or contexts. Cultural, educational, and healthcare system differences in other countries could affect the stress levels and coping mechanisms of nursing students, thus limiting the broader applicability of our results. Further research involving diverse populations and settings is necessary to validate and extend the findings of this study.

In addition, despite extensive outreach efforts through social networks, we were unable to obtain responses from all universities in Spain. This lack of representation from certain institutions may affect the generalizability of our findings. It is also worth noting that participation in the study was voluntary. This could introduce selection bias, as those who chose to participate might have different stress levels or coping mechanisms compared to those who did not participate.

Furthermore, while the sample size was large, to extrapolate the results to the entire nursing student population in Spain with greater confidence, an even larger and more representative sample would be needed. This would help to ensure that the findings are truly reflective of the broader student population and not limited to the experiences of a specific subset.

Finally, it is important to consider that the study relied on self-reported data, which can be subject to biases such as social desirability or recall bias. Despite these limitations, the study provides valuable insights into the stress levels experienced by nursing students during the COVID-19 pandemic and underscores the need for targeted interventions to support this vulnerable population.

### 4.2. Gaps of the Study

Several gaps in our study highlight areas for further research and exploration. Conceptually, the study did not investigate the underlying psychological mechanisms that contribute to stress among nursing students. Future research should explore specific factors such as coping strategies, resilience, and psychological constructs that may influence stress levels. Understanding these aspects could provide a more comprehensive view of how nursing students manage stress. 

Methodologically, although we used the PSS-10 instrument, which has demonstrated good internal consistency and reliability, additional validation techniques were not applied in this study. Future research should include Confirmatory Factor Analysis (CFA) and measurement invariance testing to ensure the robustness and applicability of the PSS-10 across different demographic groups. These methodological enhancements would strengthen the reliability and validity of the instrument and provide more detailed insights into the stress levels of nursing students. By addressing these gaps, future research can build on our findings and contribute to a deeper understanding of stress among nursing students, ultimately leading to better support and interventions.

### 4.3. Recommendations

It is crucial to emphasize the importance of equipping nursing students—and by extension, their educators—with the skills necessary to effectively manage complex, stress-inducing situations, such as those encountered during the COVID-19 pandemic. The potential recurrence of similar health emergencies, including wars and natural disasters, underscores the need for proactive training. Universities should integrate specific subjects into their curricula that focus on developing these critical skills, ensuring that both students and faculty are prepared to respond to future crises with resilience and competence.

Some studies emphasize the critical role of educators in recognizing their students’ stress levels and providing effective coping strategies to manage the inevitable stressors in nursing education. By fostering awareness and offering support, teachers can help reduce the psychological symptoms associated with perceived stress, improving students’ overall well-being. To effectively address this issue, higher education institutions must accurately diagnose the problem and develop targeted intervention projects supported by university welfare programs, in line with the Institutional Educational Project [33].

By integrating such training into educational programs and promoting teacher awareness, universities can create a supportive educational environment [37]. This approach not only prepares nursing students to navigate high-stress environments more effectively but also equips educators with the tools to support their students, leading to better mental health and academic performance across the board.

## 5. Conclusions

The majority of nursing students who conducted their practical training during the COVID-19 pandemic experienced moderate levels of perceived stress. Notably, female students reported higher stress levels compared to their male counterparts. This elevated perceived stress can result in various negative outcomes, including fear, uncertainty, mental fatigue, and burnout, potentially leading to chronic fatigue and long-term burnout in the future. The most significant factor influencing stress was related to the challenges posed by the pandemic, as reflected in the responses to item 7 of the PSS-10 scale.

It is essential to emphasize that stress management strategies are crucial not only in general but particularly during health crises and other increasingly frequent emergency situations. Adopting strategies that focus on self-awareness and emotional control techniques could be highly beneficial. Implementing these strategies for nursing students could significantly improve their quality of life. Moreover, integrating stress management into the nursing curriculum could better prepare students to face future challenges with resilience and confidence.

Overall, these findings highlight the necessity for targeted interventions to support nursing students in managing stress, especially during times of health crises. By fostering a healthier and more resilient educational environment, we can help ensure the well-being and academic success of nursing students. Additionally, these interventions should focus on gender-specific stressors to address the higher levels of stress experienced by female students. This approach will better equip nursing students to handle the demands of their profession, ultimately leading to a more robust healthcare workforce.

In conclusion, recognizing and addressing the stress experienced by nursing students during their practical training, particularly in crisis situations like the COVID-19 pandemic, are paramount. Providing resources and training on stress management can enhance their coping mechanisms, leading to improved mental health and professional performance. By prioritizing these measures, educational institutions can contribute to the development of a competent and resilient nursing workforce, capable of delivering high-quality care in any circumstance.

## Figures and Tables

**Table 1 jcm-13-04943-t001:** Students by academic year and dropout intent post-pandemic.

	Academic Year			
	First	Second	Third	Fourth
n/% of students	34 7%	127 26.1%	164 33.7%	162 33.3%
n/% of students intending to leave the bachelor’s degree in nursing	46 9.4%			

**Table 2 jcm-13-04943-t002:** Descriptive statistics for EEP-10 scale responses.

Statistics		Value
Media		20.65
Medium		21
SD		5.62
Minimum		3
Maximum		40
Percentiles	25	18
	50	21
	75	24

**Table 3 jcm-13-04943-t003:** Responses to the PSS-10 scale.

Item	Never	Rarely	Sometimes	Often	Always
	n %	n %	n %	n %	n %
1. In the last month, how often have you been upset because of something that happened unexpectedly?	59 12.1%	125 25.7%	130 26.5%	121 24.8%	52 10.7%
2. In the last month, how often have you felt that you were unable to control the important things in your life?	73 15%	161 33.1%	129 26.5%	100 20.5%	24 4.9%
3. In the last month, how often have you felt nervous and “stressed”?	29 6%	78 16%	106 21.8%	162 33.3%	112 23%
4. In the last month, how often have you felt confident about your ability to handle your personal problems?	61 12.5%	145 29.8%	136 27.9%	113 23.2%	32 6.6%
5. In the last month, how often have you felt that things were going your way?	71 15.6%	151 31%	147 30.2%	95 19.5%	23 4.7%
6. In the last month, how often have you found that you could not cope with all the things that you had to do?	37 7.6%	140 28.7%	116 23.8%	118 24.2%	76 15.6%
7. In the last month, how often have you been able to control irritations in your life?	69 14.2%	208 42.7%	132 27.1%	60 12.3%	18 3.7%
8. In the last month, how often have you felt that you were on top of things?	32 6.6%	114 23.4%	165 33.9%	127 26.1%	49 10.1%
9. In the last month, how often have you been angered because of things that were outside of your control?	71 14.6%	145 29.8%	135 27.7%	100 20.5%	36 7.4%
10. In the last month, how often have you felt difficulties were piling up so high that you could not overcome them?	83 17%	135 27.7%	140 28.7%	98 20.1%	31 6.4%

**Table 4 jcm-13-04943-t004:** Group statistics and Chi-Square test for item 7 of the PSS-10 scale with gender.

			Gender	
Scale Item 7 PSS-10 ^a^			Male	Female
Always		% (N)	5.7% (6)	3.1% (12)
Often		% (N)	21% (22)	9.9% (38)
Sometimes		% (N)	21.9% (23)	28.5% (109)
Rarely		% (N)	38.1% (40)	44% (168)
Never		% (N)	13.3% (14)	14.4% (55)
Pearson’s Chi-Square			11.640	
	Sig. asymptotic (bilateral)		0.020 *	

^a^ Grouping variable: gender; * *p*-value Chi-Square test.

**Table 5 jcm-13-04943-t005:** Group statistics and Mann–Whitney U test for the overall PSS-10 score and gender.

Global Score PSS-10	Average Range	U of Mann–Whitney	Sig. Asin. (Bilateral)
Gender ^a^			
Male	199.48	15,380.000	<0.001 *
Female	256.24		

^a^ Grouping variable: gender; * *p*-value Mann–Whitney U test.

**Table 6 jcm-13-04943-t006:** Spearman’s correlation test correlations between perceived stress scale score and age.

			Sum of Items	Age
Spearman rho	Sum of items	Correlation coefficient	1.000	0.032
		Sig. (bilateral)		0.478
		N	487	487
	Age	Correlation coefficient	0.032	1.000
		Sig. (bilateral)	0.478	
		N	487	487

## Data Availability

The data presented in this study are available on request from the corresponding author. The data are not publicly available due to privacy restrictions.

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
