# Peer review of "Navigating Perceived Stress: Experiences of Nursing Students Completing Internships during the COVID-19 Pandemic in Spain"

_jcm, 2024, doi:10.3390/jcm13164943_

Round 1

Reviewer 1 Report

Comments and Suggestions for Authors

I would like to thank the authors for the time and dedication they have invested in their work and the opportunity provided by the journal to review their manuscript. The primary reason for my concern is the delay in submitting this manuscript for review and publication (the recruitment took place between March and April 2022) and its methodological limitations. Essentially, the study relies on a single scale that ignores the context and is conducted two years after the lockdown, by which time over 68 million doses of the vaccine had already been administered. This is not discussed anywhere in the manuscript.

I share my comments on each section:

Abstract

Why is there no reference to the correlations conducted in the results?
Why do the conclusions refer to health crises when the study is conducted after the vaccination campaigns have ended?

Introduction

This section is very brief and does not provide the necessary information to understand the study's results. What is the relationship between studies conducted during COVID-19, stress, and age? What are the main stressors during clinical practice for Health Sciences students? What differences exist between the stressors of nursing students before and after the pandemic?

Materials and Methods

The study's methodology is extremely deficient. It uses only one scale to assert that it evaluates stress derived from COVID-19. What inclusion and exclusion criteria were used for the participants? How have other potential sources of stress been controlled? Why was this date chosen to recruit the sample if the introduction refers to the consequences of the lockdown after March 2020? Other studies have included more relevant information previously and at a more critical time during the pandemic to evaluate the stress of nursing students, controlling variables specifically related to COVID-19 (Ahmed et al., 2022; Aslan & Pekince, 2021).

Results

Although correctly described, the study's results lack sufficient scientific relevance.

Discussion and Conclusions

The discussion should delve deeper and include previous reviews on stress and stressors during COVID-19. The limitations overlook the methodological biases present in the study.

Aslan, H., & Pekince, H. (2021). Nursing students' views on the COVID‐19 pandemic and their perceived stress levels. Perspectives in psychiatric care, 57(2), 695-701.
Ahmed, W. A., Abdulla, Y. H. A., Alkhadher, M. A., & Alshameri, F. A. (2022). Perceived stress and coping strategies among nursing students during the COVID-19 pandemic: a systematic review. Saudi Journal of Health Systems Research, 2(3), 85-93.

Comments on the Quality of English Language

The quality of English is good enough.

Reviewer 2 Report

Comments and Suggestions for Authors

Dear Author(s),

I would like to begin by expressing my gratitude to you for giving me the opportunity to review your work on: Navigating Perceived Stress: Experiences of Nursing Students Completing Internships During the COVID-19 Pandemic in Spain. I read the article with great excitement and believe that the study has the potential to provide further insights into the impact of the COVID-19 pandemic on the perceived stress levels of undergraduate nursing students. The manuscript is well-written and structured, and the language used is simple and easy to understand. But I have some comments which need improvement that might be improve the scientific quality of the manuscript. The following recommendations are presented:

Overall general evaluation of the manuscript

·       The scientific quality of the manuscript is supported by recent empirical evidence. However, my comments is what gaps were identified in terms of conceptual, contextual and methodological related to your research area. Please make sure include these important points in your manuscript by giving one subtitle “Gaps of the study”.

·       In this study, a multi-faceted approach was used to the recruitment of subjects was used, including participation by student representatives from different universities. This probably added to the representativeness of the sample.

·       The participants were required to show informed consent before data collection. Thus, it is an ethically sound study. Approval from the Huelva Research Ethics Committee lends additional support to the fact that this piece of research adhered to ethical practices is one.

·       The use of the PSS-10 instrument with good internal consistency and reliability further adds to the credibility of measurement of perceived academic stress in this group of nursing students. However, PSS-10 measure for COVID 19 were applied in different settings. Therefore, I recommended the authors applied measurement invariance, Confirmatory factor analysis in your study seatings is my strong comment. I hope it will improve your instrument quality and checking equivalence across gender and other demographic factors included in your study.

·       The fact that it was a study about nursing students in Spain within a time period may reduce generalizability to other populations or contexts, therefore limiting the broader applicability of the results. Please put this in the limitation section.

·       The discussion section deals with a very important and relevant issue—the stress levels of nursing students in the COVID-19 pandemic. This relevance adds more importance to the findings. However, my recommendation is please adding existing scientific literature in support of additional studies that address perceived stress during COVID 19. This might strengthen your manuscript scientific quality.

Overall Evaluation

Finally, I believe your study has made a valuable contribution to the field of perceived stress levels of nursing students during COVID 19. By addressing the suggestions provided and further refining the manuscript, your work has the potential to be a significant addition to the scientific community. I encourage you to carefully consider the comments and suggestions outlined above and incorporate them into your revised manuscript.

Thank you once again for giving me the opportunity to review your article, and I wish you the best of luck with your research.

Round 2

Reviewer 1 Report

Comments and Suggestions for Authors

Thank you to authors for their consideration and amendments. Now the manuscript has considerably improved and meets the criteria for publication.